# KG-DG: KNOWLEDGE-GUIDED DOMAIN GENERALIZATION VIA NEURO-SYMBOLIC FUSION

## ABSTRACT

Domain generalization remains a critical challenge in medical imaging, where models trained on single sources often fail under real-world distribution shifts. We propose **KG-DG**, a neuro-symbolic framework for diabetic retinopathy (DR) classification that integrates vision transformers with expert-guided symbolic reasoning to enable robust generalization across unseen domains. Our approach leverages clinical lesion ontologies through structured, rule-based features and retinal vessel segmentation, fusing them with deep visual representations via a confidence-weighted integration strategy. The framework addresses both single-domain generalization (SDG) and multi-domain generalization (MDG) by minimizing the KL divergence between domain embeddings, thereby enforcing alignment of high-level clinical semantics.

Extensive experiments across four public datasets (APTOS, EyePACS, Messidor-1, Messidor-2) demonstrate significant improvements: up to a 5.2% accuracy gain in cross-domain settings and a 6% improvement over baseline ViT models. Notably, our symbolic-only model achieves a 63.67% average accuracy in MDG, while the complete neuro-symbolic integration achieves the highest accuracy compared to existing published baselines and benchmarks in challenging SDG scenarios. Ablation studies reveal that lesion-based features (84.65% accuracy) substantially outperform purely neural approaches, confirming that symbolic components act as effective regularizers beyond merely enhancing interpretability. Our findings establish neuro-symbolic integration as a promising paradigm for building clinically robust, and domain-invariant medical AI systems. **Keywords**: Domain Generalization, Neuro-Symbolic Learning, Medical Imaging, Diabetic Retinopathy, Vision Transformers, Out-of-Distribution Robustness

## 1 INTRODUCTION

Diabetic Retinopathy (DR) is a microvascular complication of Diabetes Mellitus that affects the retinal vasculature, leading to hemorrhages, microaneurysms, exudates, and cotton-wool spots which, if left untreated, can culminate in irreversible vision loss Khandelwal et al. (2023). Manual grading of fundus photographs by expert ophthalmologists remains the clinical gold standard but is both time-consuming and subject to inter-observer variability Kauppi et al. (2019). Despite the success of deep learning models—particularly Vision Transformers (ViTs)—on single-source DR datasets Dosovitskiy et al. (2021); Zhao et al. (2022), their performance suffers when confronted with domain shifts caused by variations in imaging devices, resolution settings, and patient demographics. Although Domain Generalization (DG) strategies such as Empirical Risk Minimization under the DomainBed protocol Gulrajani & Lopez-Paz (2021) offer a baseline for robustness, they often overlook the integration of structured clinical knowledge and realistic augmentation techniques that are critical for reliable cross-domain deployment.

Neuro-symbolic learning, which integrates deep learning with symbolic reasoning, has gained traction as a promising strategy to improve domain generalization in medical imaging. Deep models extract complex patterns from raw data, while symbolic components encode high-level domain knowledge and constraints, thereby effec tively guiding model behavior across varying domains. This hybrid approach can mitigate overfitting to domain-specific artifacts by enforcing consistency with known anatomical or pathological rules. For example, Han et al. introduced a neuro-symbolic framework

for spinal MRI segmentation that embeds anatomical priors into a deep adversarial graph network, resulting in better generalization and interpretability across different datasets Han et al. (2021). Similarly, Ozkan and Boix demonstrated that training across multiple imaging modalities (e.g., MRI, CT, ultrasound) significantly improves generalization to unseen domains, emphasizing the value of diverse training data and domain-aware learning strategies Ozkan & Boix (2024). These findings suggest that symbolic reasoning components can serve as a regularizing force that biases models toward clinically meaningful and domain-invariant features—thereby enabling more robust, scalable medical AI systems.

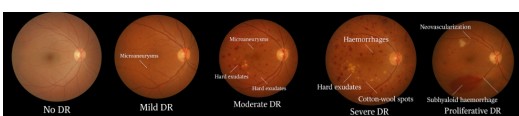

Figure 1: Fundus images showing DR progression: from No DR to Proliferative DR Kauppi et al. (2019).

Despite recent advances, most current neuro-symbolic methods remain narrowly focused—typically emphasizing symbolic reasoning mechanics without incorporating the type of clinical knowledge used by medical experts to inform robust, generalizable decision-making. In particular, few approaches simultaneously address both symbolic knowledge integration and domain generalization in a cohesive framework. This gap motivates our proposed method, **KG-DG**, a knowledge-guided domain generalization framework that unifies structured clinical knowledge with deep learning models in a scalable manner. KG-DG encodes domain-invariant biomarkers—such as exudates, hemorrhages, and vascular abnormalities—directly into the learning pipeline, guiding classification tasks while enhancing out-of-distribution (OOD) robustness.

## 1.1 DOMAIN GENERALIZATION

Vision transformers (ViTs) have revolutionized medical image analysis, particularly in ophthalmology, offering a powerful alternative to traditional convolutional neural networks. Dosovitskiy et al. Dosovitskiy et al. (2021) established the foundation by demonstrating ViTs' state-of-the-art performance on large-scale image recognition benchmarks, catalyzing their adoption for diabetic retinopathy (DR) detection. Subsequent work by Kothari et al. introduced TransDR Wang et al. (2024), enhancing ViTs with lesion-aware attention mechanisms that improve lesion localization capabilities, though without explicitly addressing domain shift robustness challenges.

In many real-world applications, particularly in biomedical fields, it is unrealistic to expect access to new patients' data before model deployment due to domain shifts between data from different patients Muandet et al. (2013). To address this challenge, the concept of Domain Generalization (DG) was introduced Blanchard et al. (2011). DG aims to train models on data from one or more related but distinct source domains, enabling them to generalize effectively to unseen, out-of-distribution (OOD) target domains. Since its formal introduction by Blanchard et al. in 2011 Blanchard et al. (2011), a wide range of techniques have been proposed to tackle the DG challenge Zhou et al. (2021)–Cha et al. (2021).

These approaches include learning domain-invariant representations by aligning source domain distributions Li et al. (2018b;d), simulating domain shifts during training using meta-learning Li et al. (2018a); Balaji et al. (2018), and generating synthetic data through domain augmentation Zhou et al. (2020b;a). From an application perspective, DG has been explored in various areas such as computer vision (e.g., object recognition Li et al. (2017; 2019), semantic segmentation Volpi & Murino (2019), and person re-identification Zhou et al. (2021; 2020b)), speech recognition Shankar et al. (2018), natural language processing Balaji et al. (2018), medical imaging Liu et al. (2020b;a), and reinforcement learning Zhou et al. (2021).

Table 1: Clinical Signs of DR Their Significance

| Symptom | Observations & Relevance |
|---|---|
| Microaneurysms | Tiny red dilations; earliest sign of Mild NPDR (Frank, 2004; Wilkinson et al., 2003). |
| Haemorrhages | Dot/blot or flame-shaped. Severe NPDR: >20 in all quadrants (of Ophthalmology, 2023; Group, 1991). |
| Hard Exudates | Lipid deposits from leakage near macula. Risk for DME (Group, 1991; Shukla & Tripathy, 2025). |
| Cotton Wool Spots | White lesions from nerve infarction. Signify ischemia (Frank, 2004; Publishing, 2024). |
| Subhyaloid Haemorrhages | Boat/D-shaped bleed. Hallmark of Proliferative DR (Yanoff & Duker, 2019; Shukla & Tripathy, 2025). |
| Neovascularization | New fragile vessels on disc (NVD) or retina (NVE). Defines PDR (Group, 1991; of Ophthalmology, 2023). |

In medical imaging, domain shift is especially prevalent due to variations across clinical sites and individual patients Liu et al. (2020a); Dou et al. (2019). Datasets like Multi-site Prostate MRI Segmentation Liu et al. (2020a) and Chest X-rays Mahajan et al. (2021) reflect this reality, with differences in imaging equipment and acquisition protocols introducing substantial distribution variability.

## 2 METHODOLOGY

We propose a general-purpose framework for *knowledge imputation into AI-based models*, enabling integration of clinically validated rules, visual biomarkers, and demographic insights into conventional learning pipelines. This approach is designed to improve **robustness**, **interpretability**, and **domain generalization**, addressing critical limitations commonly encountered in medical deployments where data heterogeneity, distribution shifts, and limited supervision can degrade model performance.

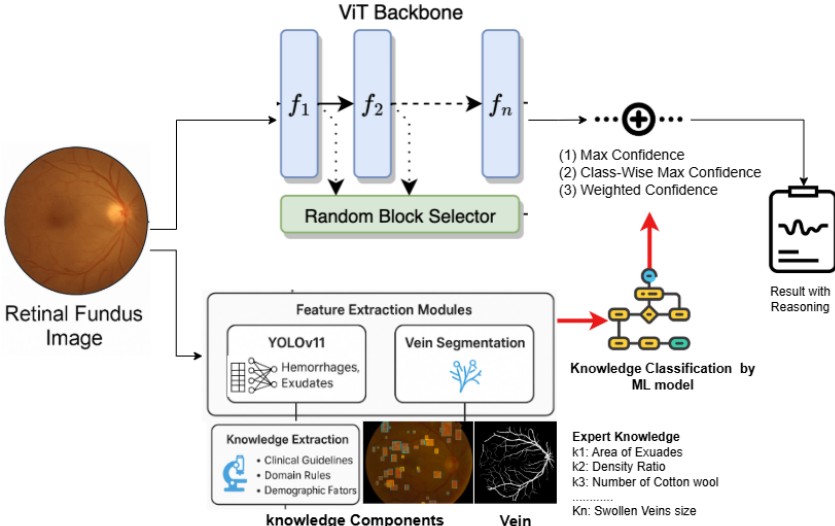

Figure 2: Overview of the proposed knowledge-guided DR classification framework, illustrating the integration of symbolic clinical rules and deep learning features Kauppi et al. (2019).

### 2.1 KNOWLEDGE-GUIDED AUGMENTATION OF DEEP MODELS

Traditional deep learning models typically learn a predictive mapping $f_{\text{DL}} : \mathcal{X} \rightarrow \mathcal{Y}$, where $\mathcal{X}$ denotes input modalities (e.g., retinal images) and $\mathcal{Y}$ represents target disease labels. This approach inherently lacks structured medical inductive biases, potentially limiting clinical applicability. To overcome this limitation, we propose a *dual-branch architecture*, integrating structured knowledge representation $\mathcal{K}$ into deep learning-based image analysis.

We formalize $\mathcal{K}$ as a set of diagnostic rules $\{r_1, r_2, \ldots, r_n\}$, each reflecting expert-validated correlations between observable clinical features and disease states. These rules incorporate visual biomarkers such as (e.g., exudates, hemorrhages, vascular patterns) and demographic parameters (e.g., patient age, glycemic status). For practical implementation, we develop corresponding *feature extractors* $C = \{c_1, c_2, \ldots, c_k\}$, instantiated via object detection models (YOLOv11), segmentation architectures, and logical rule functions.

Each extractor $c_i$ outputs a quantitative feature $f_i \in \mathbb{R}$, aggregated into a structured vector:

$$F^* = \{f_1, f_2, \ldots, f_k\}.$$

This structured vector encodes clinical attributes such as presence, severity, and spatial distribution of significant retinal lesions, facilitating symbolic reasoning aligned closely with clinical diagnostic criteria.

A parallel *knowledge-driven classifier* $f_{\text{KD}} : F^* \rightarrow \mathcal{Y}$ is trained alongside the deep learning model $f_{\text{DL}}$. The final prediction can then be determined through different fusion strategies. In the simplest case, a **selective fusion** rule is applied:

$$y_{\text{final}} = \begin{cases} y_{\text{DL}}, & \text{if } s_{\text{DL}} \geq s_{\text{KD}}, \\ y_{\text{KD}}, & \text{otherwise}, \end{cases}$$

where $s_{\text{DL}}$ and $s_{\text{KD}}$ denote the maximum confidence scores from the deep and symbolic classifiers, respectively. This strategy enhances robustness by leveraging symbolic reasoning when the deep model predictions exhibit uncertainty, particularly valuable in handling out-of-distribution scenarios. Beyond this, we experimented with three additional fusion techniques:

1. **Max Confidence Fusion**: both the neural (ViT) and symbolic classifiers output calibrated probabilities via softmax normalization. The class with the globally highest confidence is selected, irrespective of source.

2. **Class-wise Max Fusion**: normalized per-class confidence scores are compared across models, and the prediction is made according to the higher class-specific confidence.

3. **Weighted Fusion**: empirically tuned weights $(\alpha_{DL}, \alpha_{KL})$ are applied to balance neural and symbolic predictions. Formally,

$$y_{\text{final}} = \arg \max_{c \in \mathcal{C}} \left( \alpha_{DL} \cdot s_{DL}(c) + \alpha_{KL} \cdot s_{KL}(c) \right),$$

where $s_{DL}(c)$ and $s_{KL}(c)$ are the softmax confidence scores assigned by the deep and symbolic classifiers, respectively, for class $c$, and $\mathcal{C}$ is the set of all DR severity classes.

Together, these strategies allow us to assess the trade-off between model confidence, robustness, and the influence of symbolic knowledge on final decision-making.

## 2.2 DIABETIC RETINOPATHY CLASSIFICATION

We evaluated the proposed KG-DG framework on the task of diabetic retinopathy (DR) classification using retinal fundus images—well-suited for knowledge-guided learning due to the presence of clearly defined visual pathologies such as microaneurysms, hemorrhages, exudates, and neovascularization. Domain-specific diagnostic rules were curated from ophthalmological guidelines (see Table 1) and operationalized via automated feature extraction pipelines built using two open-source, modular tools: YOLOv11 and a retinal vessel segmentation model.

For lesion-level localization, we employed the YOLOv11 object detection model, a state-of-the-art one-stage detector known for its efficiency and precision in dense object environments. YOLOv11 extends the YOLOv5/YOLOv7 series with advanced improvements including CSPDarkNet-based backbones, decoupled heads, and dynamic label assignment (DLA), achieving superior mean average precision (mAP) with real-time inference capabilities Wang et al. (2022). We fine-tuned YOLOv11 to detect clinically relevant lesions such as hemorrhages, hard exudates, and cotton wool spots. Bounding boxes produced by the model were post-processed and validated using Intersection over Union (IoU) scores against expert-labeled fundus images, ensuring medical fidelity.

In parallel, we integrated a vein segmentation module to extract morphological vessel features. This module, adapted from the open-source DRIVE and CHASE-DB1 datasets, uses a modified U-Net architecture with spatial attention layers to segment retinal vessels with high sensitivity. From these segmented maps, we extracted quantitative features including vessel tortuosity, branching angles, and average caliber—biomarkers strongly associated with DR progression.

This structured knowledge vector was passed into a parallel symbolic classifier trained independently from the deep model, enabling our system to rely on rule-driven inference when the deep model exhibits uncertainty. Various machine learning models, including Logistic Regression, Random Forest, Support Vector Machines (SVM), Gradient Boosting, and K-Nearest Neighbors, were evaluated for knowledge-based classification on the feature set ($F$). Among these, Gradient Boosting demonstrated the best classification performance. Both YOLOv11 and the vein segmentation module functioned solely as independent auxiliary components to extract biomarkers from images, facilitating symbolic reasoning. The biomarkers were annotated by expert medical annotators on approximately 500

images, with random samples validated by respective domain experts. These annotations were subsequently used to fine-tune the YOLOv11 and vein segmentation modules, which then act as knowledge extractors within the pipeline. This accurate integration of clinical knowledge enhances model robustness, promotes domain invariance, and provides a solid foundation for understanding domain shifts through distributional alignment.

The classification results from the knowledge-based machine learning model and the ViT model are integrated using three main methods as shown in Figure 2: (1) selecting the maximum confidence score across all predictions, (2) computing the class-wise maximum confidence, and (3) applying a weighted confidence scheme. The outcomes of these three integration strategies are evaluated to assess the overall performance of the final framework.

## 2.3 Backbone Architectures and Training Strategy

For the image-based analysis, we employed advanced Vision Transformer (ViT) architectures. The DeiT-small architecture, comprising approximately 22M parameters, was used without distillation Touvron et al. (2021a). The CvT-13 model, with 20M parameters, integrates convolutional layers with transformer blocks to enhance spatial feature learning Wu et al. (2021). Additionally, we utilized T2T-ViT-14, featuring progressive tokenization and encompassing 21.5M parameters Yuan et al. (2021b).

All ViT models were initialized with ImageNet-pretrained weights, and during training, encoder parameters remained fixed to prevent overfitting. Only the classification heads underwent optimization using class-weighted cross-entropy loss. Training adhered to DomainBed protocols, employing resizing to 224×224, random cropping, horizontal flipping, color jitter, and grayscale augmentation. AdamW optimizer was utilized with a learning rate of $5 \times 10^{-5}$, and early stopping was implemented after 10 epochs without performance improvement.

## 2.4 Evaluation Protocol and Results

Initially, KG-DG is evaluated on the Aptos Dataset (60% training, 20% cross-validation, and 20% testing), achieving superior performance, exceeding a ViT benchmark by 6% (84.65% vs. 78.40%) and significantly outperforming existing baselines. We conducted extensive evaluations in both multi-source and single-source domain generalization settings using publicly available DR datasets: APTOS Kauppi et al. (2019), EyePACS Kaggle (2015), MESSIDOR, and MESSIDOR2 Decencière et al. (2014). Each dataset constituted a distinct domain. In multi-source experiments, we trained models on three datasets while testing on the fourth. In single-source setups, we trained on a single dataset and evaluated on the remaining domains.

Our knowledge-guided framework consistently demonstrated superior performance, achieving a +2.1% average accuracy improvement in multi-source domain generalization and a notable +4.2% increase in single-source domain generalization scenarios, particularly impactful on imbalanced data distributions (see detailed results in Table 6).

The structured knowledge-driven classifier notably improved generalization by encapsulating domain-invariant medical reasoning, whereas the deep learning branch effectively modeled intricate visual patterns, validating the effectiveness of integrating clinical expertise within deep learning frameworks.

*Note. Unless otherwise stated, in all tables the best-performing value within each column is highlighted in **bold**.*

## 3 Experiments

### 3.1 Single Domain Generalization Results

In the SDG setting, models were trained on one dataset and evaluated on the remaining three to simulate clinical deployment in unseen environments. Our method was evaluated against DRGen, ERM-ViT, SD-ViT, and SPSD-ViT using APTOS Kauppi et al. (2019), EyePACS Kaggle (2015), Messidor-1 and Messidor-2. Decencière et al. (2014) as source domains respectively. As shown in

Table 2: Single Domain Generalization (SDG) Cross-domain Accuracy (%). Models were trained on one source domain and evaluated on the three unseen target domains. The highest average accuracy is in bold.

Table 3: Trained on APTOS

| Method | Eyepacs | Messidor | Messidor2 | Avg |
|---|---|---|---|---|
| DRGen | 67.5±1.8 | 46.7±0.1 | **61.0**±0.1 | 58.4 |
| ERM-ViT | 67.8±1.4 | 45.5±0.2 | 58.8±0.4 | 57.3 |
| SD-ViT | 72.0±0.8 | 45.4±0.1 | 58.5±0.2 | 58.6 |
| SPSD-ViT | 71.4±0.8 | 45.6±0.1 | 58.8±0.2 | 58.6 |
| VIT (DL) | 66.6±0.4 | 46.4±0.3 | 48.9±0.2 | 53.9 |
| Knowledge (KL) | 66.4±0.8 | **49.6**±0.2 | 53.9±0.7 | 56.6 |
| NonW (DL+KL) | **72.8**±0.5 | 50.6±0.4 | 54.3±0.4 | **59.9** |
| Weighted | 67.4±0.3 | 49.6±0.3 | 53.9±0.6 | 57.0 |

Table 4: Trained on MESSIDOR

| Method | Aptos | Eyepacs | Messidor2 | Avg |
|---|---|---|---|---|
| DRGen | 41.7±4.3 | 43.1±7.9 | 44.8±0.9 | 43.2 |
| ERM-ViT | 45.3±1.3 | 52.4±3.2 | 58.2±3.2 | 51.9 |
| SD-ViT | 44.3±0.9 | 53.2±1.6 | 57.8±2.4 | 51.7 |
| SPSD-ViT | 48.3±1.1 | 57.4±2.1 | 62.2±1.6 | 55.9 |
| VIT (DL) | 49.8±0.4 | 62.1±0.3 | 59.1±0.3 | 57.0 |
| Knowledge (KL) | **74.0**±0.5 | 63.6±0.4 | **63.8**±0.3 | **67.1** |
| NonW (DL+KL) | 52.7±0.7 | 63.4±0.4 | 61.4±0.5 | 59.2 |
| Weighted | 74.1±0.5 | 63.3±0.2 | 63.8±0.6 | 67.1 |

Table 5: Trained on MESSIDOR2

| Method | Aptos | Eyepacs | Messidor | Avg |
|---|---|---|---|---|
| DRGen | 40.9±3.9 | 69.3±1.0 | 61.3±0.8 | 57.7 |
| ERM-ViT | 47.9±2.1 | 67.4±0.9 | 59.6±3.9 | 58.3 |
| SD-ViT | 51.8±0.9 | 68.7±0.6 | **62.0**±1.7 | 60.8 |
| SPSD-ViT | 52.8±2.0 | **72.5**±0.3 | 61.0±0.8 | 62.1 |
| VIT (DL) | 29.2±0.4 | 44.7±0.5 | 49.4±0.7 | 41.1 |
| Knowledge (KL) | **69.1**±0.3 | 71.1±0.4 | 55.3±0.9 | 65.2 |
| NonW (DL+KL) | 63.6±0.6 | 71.1±0.8 | 56.4±0.2 | 63.7 |
| Weighted | 69.5±0.4 | 71.0±0.2 | 55.9±0.6 | **65.5** |

Table 6: Trained on EYEPACS

| Method | Aptos | Messidor | Messidor2 | Avg |
|---|---|---|---|---|
| DRGen | 61.3±1.9 | 54.6±1.5 | 65.4±0.1 | 60.4 |
| ERM-ViT | 69.1±1.4 | 50.4±0.3 | 62.8±0.2 | 60.8 |
| SD-ViT | 69.3±0.3 | 50.0±0.5 | 62.9±0.2 | 60.7 |
| SPSD-ViT | **75.1**±0.5 | 50.5±0.8 | 62.2±0.4 | **62.5** |
| VIT (DL) | 49.7±0.9 | 52.9±0.2 | 49.1±0.9 | 50.6 |
| Knowledge (KL) | 60.2±0.2 | **53.7**±0.6 | 66.5±0.4 | 60.1 |
| NonW (DL+KL) | 63.9±0.2 | 53.8±0.3 | **67.2**±0.6 | 61.7 |
| Weighted | 60.2±0.3 | 48.7±0.2 | 66.4±0.7 | 58.4 |

Tables 2-5, our method consistently outperformed existing baselines in three out of four training configurations.

For instance, when trained on APTOS, the Non-Weighted DL+KL fusion achieved the highest average accuracy (59.9%), outperforming all transformer baselines and showing superior generalization to diverse domains like MESSIDOR2. Similarly, when trained on MESSIDOR2, the Weighted DL+KL fusion delivered a performance of 65.5%, highlighting robustness against shifts in both demographic and imaging characteristics. These results validate that symbolic knowledge integration enables effective generalization from a single domain, crucial for low-resource clinical settings.

## 3.2 Multi Domain Generalization Results

In the MDG setting, we trained our model on three datasets and evaluated on the unseen fourth, as per the DomainBed protocol. Results in Table 7 show that our KG-DG model using Clip-ViT (ViT+KL) and symbolic classifiers significantly improved generalization compared to popular convolutional and transformer-based DG methods, including ERM, IRM, Fishr, and SD-ViT. Notably, the knowledge-guided symbolic model (KL only) achieved the best average accuracy (63.67%), while SPSD-ViT and ERM-ViT with strong augmentations reached 65.5%. Despite having fewer parameters, our model's performance indicates effective utilization of symbolic lesion features and their generalization power across domain shifts. In particular, the KL model exceeded both standard ViT and ResNet baselines across most target domains, demonstrating the critical role of encoded clinical knowledge in cross-domain settings.

## 4 Evaluation

### 4.1 Benchmark Setup

To rigorously evaluate the generalization capability of the proposed KG-DG framework, we conducted experiments on four publicly available diabetic retinopathy (DR) fundus image datasets: APTOS Kauppi et al. (2019), EyePACS, Messidor-1, and Messidor-2. Each dataset represents a distinct clinical domain, differing significantly in patient demographics, imaging devices, and image acquisition protocols. Following the DomainBed benchmark protocol established by Gulrajani et al. Gulrajani & Lopez-Paz (2021), we implemented two experimental scenarios: Single-Domain Generalization

Table 7: Performance comparison of different methods and backbones across diabetic retinopathy datasets (Accuracy %).

| Method | Backbone (#Param) | Aptos | Eyepacs | Messidor | Messidor 2 | Avg. |
|---|---|---|---|---|---|---|
| ERM Vapnik (1999) | ResNet50$_{(23.5M)}$ | 47.6±1.7 | 71.3±0.3 | 63.0±0.4 | 69.0±1.5 | 62.7 |
| IRM Arjovsky et al. (2019) | ResNet50 | 52.1±1.7 | 73.2±0.3 | 51.3±3.8 | 57.2±1.7 | 58.4 |
| ARM Zhang et al. (2021) | ResNet50 | 45.6±1.5 | 71.7±0.5 | 62.4±1.0 | 60.0±3.4 | 59.9 |
| Fish Shi et al. (2021) | ResNet50 | 44.6±2.2 | 72.7±0.7 | 62.1±0.7 | 66.4±1.7 | 61.4 |
| Fishr Rame et al. (2022) | ResNet50 | 47.0±1.8 | 71.9±0.6 | 63.3±0.5 | 66.4±0.2 | 62.2 |
| GroupDRO Sagawa et al. (2020) | ResNet50 | 44.9±3.8 | 72.0±0.3 | 63.1±0.9 | 67.8±1.9 | 62.0 |
| MLDG Li et al. (2018a) | ResNet50 | 44.1±1.6 | 72.7±0.6 | 62.7±0.6 | 64.4±0.4 | 61.0 |
| Mixup Yan et al. (2020) | ResNet50 | 47.3±1.7 | 72.0±0.3 | 59.8±2.8 | 65.8±1.4 | 61.2 |
| Coral Sun & Saenko (2016) | ResNet50 | 49.8±1.0 | 71.7±0.9 | 58.6±2.8 | 68.2±0.6 | 62.1 |
| MMD Li et al. (2018b) | ResNet50 | 49.3±1.0 | 69.3±1.1 | 64.1±4.8 | 69.6±0.6 | 63.1 |
| DANN Ganin et al. (2016) | ResNet50 | **54.4±0.8** | 72.9±1.4 | 57.0±1.1 | 58.6±1.7 | 60.7 |
| CDANN Li et al. (2018c) | ResNet50 | 48.1±0.7 | 73.1±0.3 | 55.8±1.8 | 61.2±1.3 | 59.5 |
| ERM-ViT Vapnik (1999) | DeiT-Small$_{(22M)}$ | 48.5±0.9 | 70.7±1.7 | 62.7±1.6 | 69.5±2.5 | 62.9 |
| ERM-ViT Vapnik (1999) | T2T-14$_{(21.5M)}$ | 54.0±3.0 | 73.2±0.4 | 60.8±1.7 | 72.0±0.2 | 62.5 |
| ERM-ViT Vapnik (1999) | CvT-13$_{(20M)}$ | 51.3±1.7 | 73.3±0.2 | 64.8±0.6 | 72.4±0.6 | 65.5 |
| SD-ViT Sultana et al. (2022) | DeiT-Small$_{(22M)}$ | 48.2±2.5 | 69.6±1.5 | 63.9±1.3 | 65.0±1.7 | 61.8 |
| SD-ViT Sultana et al. (2022) | T2T-14$_{(21.5M)}$ | 46.5±0.8 | 71.1±0.7 | 63.9±0.9 | 71.4±0.2 | 63.2 |
| SPSD-ViT Jayanga et al. (2023) | DeiT-Small$_{(22M)}$ | 51.6±1.1 | 73.3±0.4 | 64.0±1.4 | **72.9±0.1** | 65.5 |
| SPSD-ViT Jayanga et al. (2023) | T2T-14$_{(21.5M)}$ | 50.0±2.8 | **73.6±0.3** | **65.2±0.3** | **73.3±0.2** | **65.5** |
| SPSD-ViT Jayanga et al. (2023) | CvT-13$_{(20M)}$ | **51.7±1.2** | 73.3±0.2 | 64.8±0.6 | 72.4±0.6 | 65.5 |
| **ViT (Ours)** | Vit $_{(22M)}$ | 50.1±1.7 | 69.4±0.3 | 58.13±3.8 | 67.1±1.7 | 61.18 |
| **ViT +KL (Ours)** | Vit$_{(21.5M)}$ | 53.1±1.7 | 72.2±0.3 | 51.3±3.8 | 56.2±1.7 | 58.4 |
| **KL (Ours)** | Knowledge$_{(20M)}$ | **60.70±1.2** | 68.45±0.2 | 58.67±0.6 | 67.66±0.6 | 63.67 |

(SDG), wherein the model is trained on a single domain and evaluated on the remaining three domains, and Multi-Domain Generalization (MDG), where training is performed on three domains with evaluation conducted on a separate unseen domain.

For preprocessing, all images were uniformly resized to $224 \times 224$ pixels and subjected to data augmentations including center cropping, horizontal flipping, color jittering, and grayscale conversion to mimic realistic variability and prevent dataset-specific biases. To ensure a robust and unbiased evaluation, early stopping was applied based on validation accuracy computed on the training domain(s).

## 4.2 BASELINE MODELS

We evaluated our KG-DG framework against several competitive baseline methods representative of both convolutional neural network (CNN)-based and transformer-based domain generalization strategies. For convolutional architectures, we included Empirical Risk Minimization (ERM) with ResNet-50 He et al. (2016), a strong baseline under fair evaluation standards Gulrajani & Lopez-Paz (2021). Additionally, we compared against Invariant Risk Minimization (IRM) Arjovsky et al. (2019), Group Distributionally Robust Optimization (GroupDRO) Sagawa et al. (2020), Fishr Rame et al. (2022), and Adaptive Risk Minimization (ARM) Zhang et al. (2021), each employing distinct strategies to enforce robustness and domain invariance.

Transformer-based models considered included ERM-ViT with DeiT-Small Touvron et al. (2021b), CvT-13 Wu et al. (2021), and T2T-ViT Yuan et al. (2021a). We further included state-of-the-art transformer-based domain generalization models, SD-ViT Sultana et al. (2022) and SPSD-ViT Jayanga et al. (2023), which utilize semantic alignment and pseudo-labeling to enhance robustness. Lastly, we compared against DRGen Atwany & Yaqub (2022), a DR-specific DG method leveraging adversarial and contrastive learning.

The evaluation of our framework and comparative methods was performed using multiple metrics designed to comprehensively assess the models' performance under domain shift. Cross-domain accuracy was employed as the primary metric to gauge generalization effectiveness on unseen datasets. To address inherent class imbalance common in diabetic retinopathy classification tasks, we reported the Macro F1-score, which provides a balanced measure across all DR severity classes. Additionally, we calculated the Area Under the Receiver Operating Characteristic Curve (AUC-ROC), offering insights into sensitivity-specificity trade-offs critical in medical diagnostics.

To quantify distributional alignment across domains, we employed KL divergence between domain-specific embeddings. Lastly, the quality and reliability of our symbolic lesion detection modules were assessed through Intersection-over-Union (IoU) scores against expert annotations, ensuring clinical relevance and interpretability of the symbolic knowledge incorporated into our framework.

### 4.3 ABLATION STUDY

**Ablation Study I: APTOS-Trained Domain Generalization** To understand the individual and combined contributions of neural and symbolic components in our framework, we conducted a focused ablation study using the APTOS dataset as the source domain. Table 8 reports the accuracy performance on three unseen target domains—EyePACS, Messidor-1 and Messidor-2 when models were trained solely on APTOS.

The neural-only baseline using Vision Transformer (ViT) achieves a modest average accuracy of 53.9%, indicating limited generalization under domain shift. The symbolic-only model, based on knowledge-driven lesion features (KL), improves the average accuracy to 56.6%, highlighting the value of structured clinical priors. The best performance is observed when combining both neural and symbolic reasoning. In particular, the non-weighted fusion approach yields the highest average accuracy of 59.9%, outperforming both standalone models. This result demonstrates the strength of the proposed neuro-symbolic integration in improving robustness and domain generalization in diabetic retinopathy classification.

Table 8: Ablation study comparing neural-only (ViT), symbolic-only (KL), and fused (DL+KL) models, trained on the APTOS dataset and evaluated on unseen domains. Neuro-symbolic fusion achieves the highest average generalization accuracy.

| Setting | Eyepacs | Messidor | Messidor2 |
|---|---|---|---|
| **Neural Only (ViT)** | 66.6 | 46.4 | 48.9 |
| **Symbolic Only (KL)** | 66.4 | 49.6 | 53.9 |
| **Neural + Symbolic (Non-Weighted)** | **72.8** | **50.6** | **54.3** |
| **Neural + Symbolic (Weighted)** | 67.4 | 49.6 | 53.9 |

Table 9: **Ablation Study on Symbolic Lesion Biomarkers with and without Retinal Vein Features.** The first section evaluates performance with lesion biomarkers alone *exudates, hard hemorrhages, soft hemorrhages, and cotton wool spots* on the APTOS dataset; the second includes additional retinal vein morphology features (e.g., tortuosity, caliber, branching angles).

| Model | Feature Set | Accuracy | F1-Score | Precision | Recall | Exudate Score | Hemorrhage Score | AUC |
|---|---|---|---|---|---|---|---|---|
| Logistic Regression | Lesions Only | 0.7732 | 0.7322 | 0.59 | 0.49 | 0.77 | 0.75 | 0.74 |
| Random Forest | Lesions Only | 0.8169 | 0.8115 | 0.82 | 0.80 | 0.80 | 0.78 | 0.81 |
| SVM | Lesions Only | 0.7814 | 0.7432 | 0.59 | 0.50 | 0.77 | 0.75 | 0.76 |
| Gradient Boosting | Lesions Only | **0.8465** | **0.8412** | **0.82** | **0.76** | **0.83** | **0.80** | **0.84** |
| K-Nearest Neighbors | Lesions Only | 0.7814 | 0.7896 | 0.63 | 0.56 | 0.78 | 0.76 | 0.77 |
| Logistic Regression | Lesions + Vein | 0.6424 | 0.6019 | 0.25 | 0.33 | 0.55 | 0.58 | 0.58 |
| Random Forest | Lesions + Vein | 0.7384 | 0.7038 | 0.55 | 0.47 | 0.71 | 0.71 | 0.70 |
| SVM | Lesions + Vein | 0.6556 | 0.6083 | 0.26 | 0.34 | 0.56 | 0.59 | 0.58 |
| Gradient Boosting | Lesions + Vein | 0.7252 | 0.7389 | 0.51 | 0.44 | 0.70 | 0.70 | 0.69 |
| K-Nearest Neighbors | Lesions + Vein | 0.6987 | 0.6369 | 0.43 | 0.44 | 0.65 | 0.67 | 0.66 |

**Ablation Study II: Performance of Symbolic Lesion Biomarkers with and without Retinal Vein Features.** This experiment evaluates the discriminative capacity of structured symbolic features extracted from retinal images, focusing on four clinically validated lesion types: *exudates, hard hemorrhages, soft hemorrhages, and cotton wool spots*. The first group of results in Table 9 includes only lesion-based features, while the second incorporates additional vascular information derived from retinal vein morphology—such as tortuosity, caliber, and branching angles.

Across all classifiers, models trained solely on lesion features consistently outperform those that include both lesions and vein information. Gradient Boosting achieves the highest accuracy (84.65%) and macro F1-score (84.12%), confirming the strong discriminative value of lesion-level biomarkers. In contrast, the addition of vein-based features leads to performance degradation, indicating that vessel morphology introduces domain-sensitive variability that hampers generalization.

Accordingly, our main KG-DG framework prioritizes lesion biomarkers as the most reliable symbolic inputs, while vein features are treated as optional. This design choice also strengthens interpretability: lesion counts and distributions directly align with established clinical diagnostic protocols, whereas vessel morphology requires context-specific calibration and exhibits less transferability across domains.

### 4.4 DISCUSSION AND LIMITATIONS

The KG-DG framework achieves consistent generalization across unseen domains by combining symbolic clinical knowledge with deep visual features, though several limitations remain. Its reliance on accurate lesion-level annotations and pre-trained modules like YOLOv11 and retinal vein segmentation introduces dependency on expert-verified data, which may not be available for other medical imaging tasks. The symbolic classifier may miss complex visual cues, and fusion performance varies with confidence-weighting strategies, highlighting a need for more adaptive mechanisms. Across SDG and MDG tasks, maximum cross-domain improvement reaches 5.2% (Messidor2 $\rightarrow$ APTOS), with average gains around 2–3%, indicating steady but not uniform dominance. Feature importance analysis shows lesion features such as "exudates count" and "hemorrhage density" align with clinical practice, supporting human-aligned decision-making. Future work could explore dynamic neuro-symbolic reasoning, integrate temporal clinical data, and extend KG-DG to other modalities like OCT or histopathology.

## 5 CONCLUSIONS

This paper introduces KG-DG, an improved knowledge-guided domain generalization framework specifically tailored for medical imaging applications, as exemplified by diabetic retinopathy classification. KG-DG integrates symbolic clinical reasoning and deep visual representations through a confidence-weighted fusion approach, significantly enhancing robustness and interpretability. Comprehensive experimental results on four diverse DR datasets demonstrated that KG-DG consistently achieved superior performance compared to strong baselines of domain generalization methods, achieving notable improvements in both single-source and multi-source generalization settings, with gains of up to 5.2% accuracy in cross-domain accuracy.

Our findings underscore the importance of embedding structured clinical knowledge within deep learning models, thereby significantly improving generalization and trustworthiness in clinical settings. Future directions include adapting the KG-DG framework to additional medical imaging modalities, such as optical coherence tomography and histopathology, and further integrating dynamic symbolic reasoning via neuro-symbolic architectures, enhancing real-time decision support capabilities in medical AI deployments. **Insights:** Our observations indicate that the integration of symbolic clinical knowledge into traditional architectures—whether Vision Transformers (ViTs) or domain-specific models such as DeepXSOZ Shama et al. (2023)—consistently leads to significant improvements in classification accuracy. Furthermore, this knowledge imputation enhances both domain generalization and the explainability of model behavior, addressing critical challenges in clinical deployment.

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
