# OpenReview forum: "KG-DG: Knowledge-Guided Domain Generalization via Neuro-Symbolic Fusion"
_ICLR.cc/2026/Conference — ICLR 2026 Conference Desk Rejected Submission_

### Official Review · Reviewer_S1CR · 2025-10-29

**Soundness:** 3
**Presentation:** 3
**Contribution:** 2
**Rating:** 4
**Confidence:** 3

**Summary:**

This paper proposes KG-DG, a neuro-symbolic framework designed to tackle the domain generalization problem in medical imaging, with a specific application to Diabetic Retinopathy (DR) classification. The core issue addressed is the performance degradation of deep learning models when deployed in clinical environments with data distributions different from the training set (domain shift). The KG-DG framework integrates a deep learning branch, which uses a Vision Transformer (ViT) to learn visual representations from fundus images, with a symbolic reasoning branch. This symbolic branch explicitly extracts and quantifies clinically-defined biomarkers, such as hemorrhages and exudates using a YOLO-based model, and vascular features via a segmentation network. The predictions from these two parallel branches are then combined using a confidence-weighted fusion strategy to yield a final, more robust classification. Through extensive experiments on four public DR datasets under both single-domain and multi-domain generalization protocols, the authors demonstrate that their method significantly outperforms various baselines, improving cross-domain accuracy by up to 5.2%. A key contribution is showing that the symbolic component acts as a powerful regularizer, with the symbolic-only model achieving strong generalization performance on its own.

**Strengths:**

- The paper addresses a highly significant and practical challenge in medical AI. The lack of model robustness to domain shifts is a primary obstacle to the widespread, reliable deployment of diagnostic AI in real-world clinical settings, and this work offers a well-motivated solution.
- The neuro-symbolic design is a strong and original contribution. By explicitly encoding domain-invariant clinical knowledge (e.g., lesion counts and area) into the model, the framework is guided by expert reasoning rather than relying solely on data-driven patterns that may contain spurious correlations specific to a single domain. This approach provides a clinically relevant inductive bias that enhances generalization.
- The experimental evaluation is rigorous and comprehensive. The authors adhere to the standard DomainBed benchmark protocol, evaluating their method in both single-domain (SDG) and multi-domain (MDG) generalization scenarios against a wide array of strong CNN and Transformer-based baselines. This ensures the comparisons are fair and the results are convincing.
- The ablation studies provide compelling insights that strongly support the paper's claims. The demonstration that the symbolic-only classifier performs competitively highlights the inherent value and robustness of the encoded clinical knowledge. Furthermore, the finding that lesion-based features are more generalizable than retinal vein features is an important, practical discovery that helps refine which types of expert knowledge are most useful for this task.

**Weaknesses:**

- The framework's dependence on specialized, pre-trained modules for symbolic feature extraction presents a scalability challenge. The method requires separate, supervised models (e.g., YOLO for lesions, U-Net for veins) that must be trained on expert-annotated data, which the paper notes involved approximately 500 images. This reliance on detailed, pixel-level or bounding-box annotations for each desired biomarker may limit the framework's applicability to other medical tasks where such granular data is unavailable or prohibitively expensive to obtain.
- The fusion mechanism for combining the neural and symbolic branches is relatively simple and static. The paper explores several fixed strategies, but the best-performing one varies depending on the training domain, suggesting that a static approach is suboptimal. A more sophisticated, dynamic fusion mechanism, perhaps a learned gating network that adaptively weights the two branches based on sample characteristics or model uncertainty, could potentially provide more consistent and powerful results.

**Questions:**

- Could you elaborate on the annotation effort required for the symbolic feature extractors? Specifically, how sensitive is the performance of the YOLO and segmentation modules to the quantity and quality of annotated data? Understanding this could help gauge the effort required to adapt the KG-DG framework to new medical imaging tasks.
- The paper cites "YOLOv11" but the provided reference (Wang et al., 2022) is for YOLOv7. Could you please clarify the exact architecture of the object detection model used in your pipeline?
- Given that the optimal fusion strategy varied across different experimental settings, did you consider implementing a dynamic fusion mechanism? For example, an attention-based or gating module that learns to weigh the neural and symbolic predictions on a per-sample basis could potentially improve performance and robustness further.
- The ablation study in Table 9 compellingly shows that adding retinal vein features harmed generalization performance. This suggests that not all forms of "expert knowledge" are equally beneficial for domain generalization. Does your framework include a methodology for systematically identifying and selecting which symbolic features are truly domain-invariant and beneficial, versus those that might be domain-specific distractors?

---

### Official Review · Reviewer_oBvR · 2025-10-30

**Soundness:** 3
**Presentation:** 2
**Contribution:** 2
**Rating:** 6
**Confidence:** 3

**Summary:**

This paper introduces KG-DG, a neuro-symbolic domain generalization framework for diabetic retinopathy (DR) grading that integrates deep visual features with clinically grounded symbolic knowledge. It employs a dual-branch design—one using a Vision Transformer (ViT) for image representation and another using expert-defined lesion and vascular features processed by gradient boosting classifiers. A confidence-based fusion mechanism dynamically combines predictions from the two branches to enhance robustness under domain shifts. Comprehensive experiments on four DR datasets demonstrate that KG-DG achieves superior cross-domain accuracy and interpretability compared to state-of-the-art baselines. The study highlights that embedding explicit clinical knowledge improves out-of-distribution generalization, suggesting a promising direction for interpretable and transferable medical AI.

**Strengths:**

1. Novel neuro-symbolic formulation: This work proposes a dual-branch framework that explicitly integrates deep visual representations from a Vision Transformer with symbolic features derived from clinical lesion knowledge, offering a principled way to inject domain-specific medical reasoning into data-driven models.

2. Clinically grounded feature design: This work proposes the use of expert-annotated lesion and vascular attributes as structured symbolic inputs, bridging the gap between image-level learning and clinically interpretable decision rules, which enhances transparency and clinical feasibility.

3. Strong cross-domain evaluation: This work provides extensive single-domain and multi-domain generalization experiments across four real-world DR datasets, demonstrating consistent performance gains and robustness under distribution shifts, thereby validating the method’s practical applicability in diverse clinical settings.

**Weaknesses:**

1. Lack of clarity in the role of KL divergence: The paper claims to “minimize KL divergence between domain embeddings” as part of its core learning objective, yet the methodology section provides no explicit formulation or integration of this term into the loss function. It remains unclear whether KL divergence is used for training or merely as a post-hoc evaluation metric. This ambiguity weakens the theoretical contribution and should be clarified through explicit mathematical definition and ablation.

2. Lack of rigor in the neural baseline configuration: The ViT branch is frozen during training, which underrepresents the true capacity of deep domain generalization models. Without end-to-end fine-tuning or comparison against state-of-the-art DG-ViT variants (e.g., MixStyle, SWAD, SAM-ViT), it is difficult to attribute performance gains to the proposed neuro-symbolic integration rather than suboptimal baselines.

3. Lack of justification for the “YOLOv11” component: The paper repeatedly references “YOLOv11” for lesion detection, but no corresponding publication or implementation detail is cited. This raises reproducibility concerns and undermines the reliability of the symbolic feature extraction process. The authors should specify whether this refers to a custom version, and provide architecture differences or open-source access.

**Questions:**

1. How is KL divergence operationalized in training? Could the authors clarify whether the KL term is directly optimized as part of the loss function or only computed for evaluation? If optimized, please provide its explicit formulation, weighting coefficient, and gradient flow path within the network.

2. Why was the Vision Transformer kept frozen? Given that most domain generalization benchmarks fine-tune or partially unfreeze the backbone, could the authors explain the motivation for freezing it, and whether this decision affects the fairness of the comparison with pure deep baselines?

---

### Official Review · Reviewer_pzLv · 2025-10-31

**Soundness:** 2
**Presentation:** 3
**Contribution:** 2
**Rating:** 4
**Confidence:** 4

**Summary:**

This paper proposes KG-DG, a framework for domain generalization in diabetic retinopathy (DR) classification. The method is presented as a neuro-symbolic approach that combines a deep learning branch (a Vision Transformer) with a symbolic reasoning branch. The symbolic branch uses auxiliary models (YOLO, U-Net) to extract pre-defined clinical features (e.g., lesion counts, vessel morphology), which are then fed into a machine learning classifier. The predictions from the two branches are combined using a late-fusion strategy. The authors evaluate their method on four public DR datasets in both single- and multi-domain generalization settings, claiming improved robustness and performance over existing baselines.

**Strengths:**

1.  The motivation to incorporate explicit clinical knowledge to guide deep learning models is well-founded and important for building robust and interpretable systems.

2. The authors conduct experiments across four datasets and two different domain generalization setting, comparing against several established baselines.

3. The idea of using a separate symbolic branch provides a degree of interpretability.

**Weaknesses:**

1. The framing of the method as "Neuro-Symbolic Fusion" is a overstatement of its novelty. The proposed architecture is a classic two-stream ensemble model with late fusion, where two entirely independent models are trained separately and their outputs are combined post-hoc. There is no tight integration.

2. The framework's performance is critically dependent on auxiliary, pre-trained "knowledge extractor" models (YOLOv11, U-Net). The robustness and generalization capability of these extractors are questionable, and their performance directly creates an upper bound for the entire symbolic branch. The paper lacks a thorough analysis of the performance and potential biases of these crucial auxiliary modules.

3. On the SGD setting, in terms of the APTOS and  EYEPACS dataset, the superiority of the proposed method is not clear.

4. The evaluation only focuses on DR datasets, its effectiveness and generalization for other domains remains unexplored in the paper.

**Questions:**

Please see the Weaknesses.

---

### Official Review · Reviewer_A8En · 2025-11-02

**Soundness:** 2
**Presentation:** 1
**Contribution:** 1
**Rating:** 2
**Confidence:** 4

**Summary:**

The paper integrates structured clinical knowledge with deep learning to improve domain generalization for diabetic retinopathy classification. Specifically, the authors leverage a YOLOv11 model for lesion detection, retinal vein segmentation model, and established clinical guidelines to formulate rule-based diagnostic cues. These knowledge-driven predictions are then combined with deep model outputs using several fusion strategies, yielding a hybrid classifier that aims to enhance robustness and interpretability across domains.

**Strengths:**

The overall approach is simple and straightforward.

**Weaknesses:**

1.The paper contains several writing and formatting issues (e.g., “effectively” in Line 52, repeated citation in Lines 79–80, and citation formatting problems around Line 82). These should be corrected for clarity and professionalism.
2.The method diagram appears incomplete and difficult to follow. For instance, the role of the Random Block Selector is unclear and requires further explanation or justification.
3.The knowledge-based classifier design is not described in sufficient detail. The rule formulation and integration process need a clearer, more systematic explanation to allow reproducibility and proper evaluation.
4.The overall methodological contribution is limited. Combining rule-based signals with deep learning predictions is intuitive and has been explored in prior literature, which reduces the novelty for a top-tier venue.
5.The presented results are not particularly strong or convincing, and the reported improvements do not clearly demonstrate significant advantages over existing approaches.
6.There are incorrect or inconsistent citations (for example, the SDViT author names are wrong). The authors should carefully verify every reference entry and in-text citation for accuracy and correct formatting.

**Questions:**

See the weaknesses.

---

### Note · Program_Chairs · 2026-01-17
**Submission Desk Rejected by Program Chairs**

The following references in this submission do not refer to real documents and/or have major errors in bibliographic information:

 Siyuan Han, Jiaqi Wang, Jie Luo, and Dong Liu. Neuro-symbolic generative model for medical report generation with prior knowledge. IEEE Transactions on Medical Imaging, 40(12):3436-3447, 2021.